# Mutual Information Preserving Neural Network Pruning

## Abstract

Model pruning is attracting increasing interest because of its positive implications in terms of resource consumption and costs. A variety of methods have been developed in the past years. In particular, structured pruning techniques discern the importance of nodes in neural networks (NNs) and filters in convolutional neural networks (CNNs). Global versions of these rank all nodes in a network and select the top-$k$, offering an advantage over local methods that rank nodes only within individual layers. By evaluating all nodes simultaneously, global techniques provide greater control over the network architecture, which improves performance. However, the ranking and selecting process carried out during global pruning can have several major drawbacks. First, the ranking is not updated in real time based on the pruning already performed, making it unable to account for inter-node interactions. Second, it is not uncommon for whole layers to be removed from a model, which leads to untrainable networks. Lastly, global pruning methods do not offer any guarantees regarding re-training. In order to address these issues, we introduce Mutual Information Preserving Pruning (MIPP). The fundamental principle of our method is to select nodes such that the mutual information (MI) between the activations of adjacent layers is maintained. We evaluate MIPP on an array of vision models and datasets, including a pre-trained ResNet50 on ImageNet, where we demonstrate MIPP's ability to outperform state-of-the-art methods. The implementation of MIPP will be made available upon publication.

## 1 Introduction

It is well-established that to limit a model's resource requirements while maintaining its accuracy, it is preferable to *prune* and re-train a large model of high accuracy rather than train a smaller model from the beginning (LeCun et al., 1989; 1998; Li et al., 2017; Han et al., 2015). Pruning can be categorized into unstructured (LeCun et al., 1989; Han et al., 2015; Li et al., 2017; Singh & Alistarh, 2020) and structured (Li et al., 2017; Zhang et al., 2021; Wang et al., 2020; Wang & Fu, 2023). Unstructured pruning selects individual weights to retain; while this offers maximum control it produces models that are not hardware-compatible and can only be deployed as sparse matrices (Han et al., 2015; Wen et al., 2016). Structured pruning, on the other hand, typically involves pruning nodes in multilayer perceptrons (MLPs) or filters in convolutional neural networks (CNNs). Unlike unstructured pruning, structured approaches generate neural networks (NNs) that can be compactly stored at the time of deployment, thereby reducing resource consumption.

Research into structured pruning methods can be categorized into two complementary approaches. One focuses on enhancing the method used to determine node importance (LeCun et al., 1998; Hassibi & Stork, 1992; Han et al., 2016; Li et al., 2017; Nonnenmacher et al., 2022), while the other aims to refine the regularization technique used to reduce the value of the pruned nodes activations to zero (Wang et al., 2020; Zhang et al., 2021; Wang et al., 2021; Wang & Fu, 2023). Generally, existing methods of node selection require that the nodes are ranked, and then the top-$k$ are maintained while the remainder are pruned (Wang et al., 2022). These steps can be carried out globally or locally. The former involves ranking all nodes across all layers (Liu et al., 2017; Wang et al., 2019), whereas local methods only consider a given layer (Zhao et al., 2019; Sung et al., 2024). Global methods are preferred because they allow control over the neural architecture, thereby improving performance (Blalock et al., 2020); however, this control over the architecture is not devoid of issues. Namely, entire layers can get pruned, creating untrainable bottlenecks. Additionally, simply ranking and

selecting the top-$k$ nodes, whether locally or globally, fails to consider the impact of pruning on the relative importance of the remaining nodes. Inspired by the success of iterative magnitude pruning (IMP) (Frankle & Carbin, 2019), SynFlow, an unstructured pruning method, adopted an iterative approach that efficiently resolved these issues simultaneously (Tanaka et al., 2020). In contrast, structured solutions require multiple re-training iterations, making them computationally impractical for large models (Liebenwein et al., 2020).

In this paper, we introduce Mutual Information Preserving Pruning (MIPP), a structured activation-based pruning technique. MIPP ensures that the mutual information (MI) shared between activations in adjacent layers is preserved during pruning. Rather than ranking nodes and selecting the top-$k$, MIPP uses the transfer entropy redundancy critereon (TERC) to dynamically prune nodes whose activations do not transfer entropy to the downstream layer (Westphal et al., 2024). Pruning in this fashion affords MIPP the following major advantages: first, maintaining the MI between the activations in adjacent layers ensures that there exists a function such that the activations of the downstream layer can be approximated using those of the pruned upstream layer, thus preserving re-trainability. Second, MIPP has the ability to consider not only long-range and local interactions but can also dynamically update these considerations in real-time depending on the nodes that have been pruned. Finally, using this dynamic method of node selection, we maintain maximum control over the network structure, preventing the rigid structure associated with local pruning and the vanishing layers associated with global techniques. To summarize, the contributions of this work are as follows:

- We develop MIPP, an activation-based pruning method that preserves MI between the activations of adjacent layers in a deep NN. We prove that perfect MI preservation ensures the existence of a function, discoverable by gradient descent, that can approximate the activations of the downstream layer from the activations of the preceding pruned layer. Consequently, MIPP implies re-trainability.

- We show that MIPP only selects nodes if they transfer entropy to the subsequent layer. This dynamic method of node selection natively considers long- and short-range interactions, while concurrently establishing per-layer pruning ratios (PRs) that avoid layer collapse.

- Through comprehensive experimental evaluation, we demonstrate that MIPP can effectively prune networks, whether they are trained or not.

## 2 RELATED WORK

MIPP is a structured, activation-based pruning method that is resistant to layer collapse. In the evaluation of MIPP, we aim to compare our approach to state-of-the-art structured pruning techniques, as well as to algorithms specifically designed to avoid layer collapse that are not structured. Consequently, we also review research dedicated to applying unstructured pruning techniques in a structured manner.

**Structured activation-based pruning.** Activation-based pruning methods commonly view the activations as *features* and the outputs as targets, before ranking and selecting the top-$k$ nodes in a global or local manner (He et al., 2017; Lin et al., 2020; Sui et al., 2021; He et al., 2017; Liu et al., 2018). Rather than considering the outputs as the target, some methods reconstruct the activations of the following layer from the preceding layer (Ding et al., 2019; Lin et al., 2017). The advantage of this is that the function that generates layer $l + 1$ from layer $l$ can be approximated using fewer parameters than that which generates the outputs from layer $l$. One such method, ThiNet, greedily selects nodes if they minimize the error in reconstructing the activations of the next layer (Luo et al., 2017). Adding nodes in this fashion will prevent the model's performance from degrading; however, the condition for removal is too restrictive, as it does not consider the effects of re-training. Furthermore, unlike MIPP, ThiNet is unable to establish layer-wise PRs. Liebenwein et al. (2020) developed an activation-based pruning scheme with the ability to establish layer-wise PRs. However, this method is not well adopted as it employs prohibitively expensive iterative re-training.

**Establishing layer-wise pruning ratios.** When pruning globally, the fraction of nodes removed from each layer is rarely consistent. This updates the network structure, which has been shown to improve performance (Blalock et al., 2020). However, at higher levels of sparsity, many methods experience layer collapse, resulting in an untrainable network (Lee et al., 2019; 2020). In Tanaka et al. (2020), the authors hypothesized that the iterative nature of IMP in Frankle & Carbin (2019) prevented layer

collapse. Building upon these foundations, they developed SynFlow, a computationally efficient iterative pruning technique that is known to avoid layer collapse. However, SynFlow is data-independent, which, while improving its generalizability, can lead to a reduction in performance. Tanaka et al. (2020) demonstrated that GraSP (Wang et al., 2022) was also resistant to layer collapse. Unlike SynFlow, it is data-dependent, making it a more effective pruning method, outperforming classic techniques such as SNIP (Lee et al., 2019).

**From unstructured to structured pruning.** In structured pruning, the aim is to prune nodes or filters rather than all trainable parameters (LeCun et al., 1989; Frankle & Carbin, 2019). The simplest method to convert from unstructured to structured is to average the importance assigned to all the weights associated with a given node. However, this may lead to a loss in information, particularly as influential weights can be both highly positive and highly negative. As a result, research has aimed to define functions that combine weight importances in a minimally lossy manner. In particular, the L1- and L2-norms - related to the euclidean distance - lead to minimal information loss and have proven effective for structured magnitude pruning (Han et al., 2015; Li et al., 2017; Wang & Fu, 2023). Magnitude-based pruning, while effective, lacks rigor: it does not account for long-range interactions, information redundancy, and so on. That said, the information preserving functions, such as L1- and L2-norm, are agnostic to the measure of weight importance used and have also successfully been applied to weight gradients (LeCun et al., 1998; Molchanov et al., 2017), and Hessian matrices (Hassibi & Stork, 1992; Peng et al., 2019; Wang et al., 2019; Nonnenmacher et al., 2022). For instance, SOSP ranks nodes based on an L1-normalized combination of both the first- and second-order derivatives of the weights with respect to the loss. This method has produced state-of-the-art structured pruning results, although we will demonstrate that SOSP is prone to layer-collapse at high levels of sparsity.

## 3 MUTUAL INFORMATION PRESERVING PRUNING AT A GLANCE

NNs can be represented as nested functions. More formally, if the input to a NN is given as $\boldsymbol{x}_0$, and we use $f_l$ to represent the function of the $l$-th layer, then the output tensor can be derived as follows: $\boldsymbol{x}_L^n = (f_L \circ f_{L-1} \circ f_{L-2} \circ \dots f_0)(\boldsymbol{x}_0^n) = F(\boldsymbol{x}_0^n)$. In addition, the function $f_l$ for layer can be described by: $f_l(\boldsymbol{x}_l^n) = \boldsymbol{x}_{l+1}^m = a(\boldsymbol{W}_l^{m \times n} \boldsymbol{x}_l^n + \boldsymbol{b}_l^m)$. In the above, a is an activation function, $\boldsymbol{W}_l^{m \times n}$ is a weight matrix and $\boldsymbol{x}_l^n$ is the input to that layer (LeCun et al., 1998; Goodfellow et al., 2016).

Structured pruning is the process of discovering binary mask vectors ($\boldsymbol{m}_l^n$), associated with each layer, $l$, that zero out weight matrix elements corresponding to a node or filter index. Under such circumstances the pruned layer function can be written: $f'_l(\boldsymbol{x}_l^n) = \boldsymbol{x}_{l+1}'^m = a(\boldsymbol{W}_l^{m \times n} \boldsymbol{x}_l^n \boldsymbol{m}_l^n + \boldsymbol{b}_l^m)$. We will use prime $'$ to indicate a pruned layer (Fahlman & Lebiere, 1990). By randomly sampling from the space of possible inputs and applying the function described by the NN, we realize not only the inputs as random variables but also all subsequent activations. We define $X_l^i$ as the random variable associated with the activations of node $i$ in layer $l$. Meanwhile, the set $\mathcal{X}_l = \{X_l^0, X_l^1 \dots X_l^N\}$ contains a random variable for all of the $N$ neurons in layer $l$. If a pruning mask is incorporated into the weights, the activations associated with pruned nodes remain zero, which can otherwise be seen as information theoretically null. We denote the set associated with a pruned layer as $\mathcal{X}_l'$.

We propose MIPP, a method that aims to preserve the MI between adjacent layers for all layers in a network, while maximizing sparsity. To do this, we aim to isolate masks $\boldsymbol{m}_l^n$, which, as previously mentioned, combine with the weights to produce updated layers that have certain activations equal to zero. These null activations should not lead to a reduction in the MI between the activations of these adjacent layers. More formally, this can be expressed as follows: $\mathcal{M} = \{\boldsymbol{m}_l^n \forall l \in [0, L-1] : I(\mathcal{X}_l'; \mathcal{X}_{l+1}) = I(\mathcal{X}_l; \mathcal{X}_{l+1})\}$.

## 4 MUTUAL INFORMATION PRESERVING PRUNING

In this section, we will introduce MIPP, by explaining first how isolating the masks defined in Section 3 preserve re-trainability. Then, we will discuss TERC with MI ordering, a method that selects features if they transfer entropy to the target. To follow, we will illustrate how we estimate the MI in high-dimensional spaces. We will then describe how it is possible to use TERC to preserve MI

between a pair of adjacent layers. Having discussed the MI process for two layers, we will generalize the proposed solution to the whole network.

## 4.1 MOTIVATION

We consider one-shot pruning with retraining: the objective is to reduce the number of nodes of the NN such that, after retraining, the pruned NN will achieve the same performance as the original. We will now argue that one way to achieve this would be to select a subset of nodes from each layer in such a way that there exists a function which, when applied to this subset, can still reconstruct the activations of the subsequent layer. We will then prove that the existence of this function preserves the MI between the activations of these layers.

To illustrate this, we guide the reader through the following example. Consider the case in which we generate the expected outputs of our NN from the activations of the last layer. More formally, we write $\mathcal{X}_L = textup f_{L-1}(\mathcal{X}_{L-1})$. We now wish to prune the activations preceding the outputs. This entails minimizing the number of nodes, or the cardinality of the set $\mathcal{X}'_{L-1}$, in such a manner that there exists a function that can reliably re-form $\mathcal{X}_L$. Furthermore, this function should be discoverable by gradient ascent. More formally, we would like to derive $\mathcal{X}'_{L-1}$ such that $\mathcal{X}_L = \sup_{g \in \mathcal{F}} g(\mathcal{X}'_{L-1})$. While this formulation reveals little in the way of a potential pruning operation, using the following theorem, we relate it to the MI-based objective presented in Section 3.

**Theorem 1:** *There exists a function* g *such that the activations of the subsequent layer can be re-formed from the pruned layer iff the MI between these two layers is not affected by pruning. More formally:* $\mathcal{X}_L = \sup_{g \in \mathcal{F}} g(\mathcal{X}'_{L-1}) \Leftrightarrow I(\mathcal{X}'_{L-1}; \mathcal{X}_L) = I(\mathcal{X}_{L-1}; \mathcal{X}_L)$.

*Proof.* See Appendix C.

Consequently, in this work we aim to select a set of masks ($\mathcal{M}$) that increase sparsity while preserving MI between layers. This ensures that, for each pruned layer, there exists a function, discoverable by gradient descent, that effectively reconstructs the activations of the subsequent layer using those of the pruned layer. Therefore, MIPP ensures re-trainability in a manner that is more rigorous than competing techniques for node-importance assignment.

## 4.2 PRELIMINARIES

### 4.2.1 TRANSFER ENTROPY REDUNDANCY CRITERION WITH MI ORDERING

Before describing the practical method, we now provide a summary of TERC and its application to pruning, through the incorporation of an additional step for MI-based ordering.

**TERC.** As stated in Section 3, we aim to preserve the MI between the layers in our network. The problem of MI preservation is one well-studied in the feature selection community (Battiti, 1994; Peng et al., 2005; Gao et al., 2016). Thus, we are able to deploy an out-of-the-box solution. In particular we use TERC, as not only does it preserve the MI with the target, but its temporal complexity is also linear in time with respect to the number of features (Westphal et al., 2024), a key property when working in highly dimensional feature spaces. In our case, rather than selecting features to describe a target, we are selecting nodes that best describe the following layer. Within this context, TERC can be summarized as follows: to begin, all nodes in the layer are assumed to be useful (and added to the non-pruned set). We then sequentially evaluate whether the reduction in uncertainty of the subsequent layer's activations is greater when a specific node is included in the un-pruned set rather than excluded. More formally, for a node, $X_l^i$, to remain in the set of un-pruned nodes, it must satisfy the following condition: $I(\mathcal{X}_l; \mathcal{X}_{l+1}) - I(\mathcal{X}_l \backslash X_l^i; \mathcal{X}_{l+1}) > 0$. Otherwise, it is pruned. This process is sequentially repeated for all nodes in the layer. As shown in Westphal et al. (2024), this simple technique will preserve the MI between layers.

**MI Ordering.** Before applying TERC, we sort the nodes in the pruning layer in descending order of MI with the target. For further clarification, please see Algorithm 2 in Appendix B. This adjustment ensures that we check last whether the more informative nodes transfer entropy to the activations of subsequent layers. This makes it less likely that they will be erroneously removed during the early stages of TERC when their information can be represented on aggregate by the large number of nodes still remaining in the un-pruned set.

### 4.2.2 MUTUAL INFORMATION ESTIMATION

Unless restricting oneself to scenarios inapplicable to real-world data (e.g. discrete random variables), verifying the condition in Section 4.2.1 is computationally intractable. Consequently, we must estimate whether the condition is verified by estimating the MI, for which many methods have been developed (Moon et al., 1995; Paninski, 2003; Belghazi et al., 2018; van den Oord et al., 2019; Poole et al., 2019).

For the purposes of pruning, our MI estimates need to only be considered for comparisons. Rather than a method that gives highly accurate estimates slowly (Franzese et al., 2024), we require one that emphasizes consistency and speed. For these reasons, we adopt the technique presented in Covert et al. (2020), in which the authors demonstrate that the MI between two random processes ($X$ and $Y$) can be approximated as the reduction in error estimation caused by using $X$ to predict $Y$. More formally: $I(X;Y) \approx \mathbb{E}[l(f(\emptyset), Y)] - \mathbb{E}[l(f(X), Y)]$, where f is some function approximated via loss l. If the variables are discrete, and a cross entropy loss is used, then this value is exactly equal to the ground truth MI (Gadgil et al., 2024). Even if the variables are continuous and a mean squared error loss is used, the above value approaches the MI under certain circumstances (Covert et al., 2020). To approximate the condition described in Section 4.2.1, we estimate all MIs five times before calculating confidence intervals. We then only keep nodes for which we are more than $x\%$ sure, that they transfer entropy to the subsequent layer ($I(\mathcal{X}_l; \mathcal{X}_{l+1}) > I(\mathcal{X}_l \backslash X_l^i; \mathcal{X}_{l+1})$). The value of $x\%$ naturally becomes the hyper-parameter we tune to affect the PR. For example, if $x\%$ is low, 50%, one only needs to be 50% sure that $I(\mathcal{X}_l; \mathcal{X}_{l+1}) > I(\mathcal{X}_l \backslash X_l^i; \mathcal{X}_{l+1})$, and thus, we prune sparingly. On the contrary, if it is high (for example, $x = 99\%$), we prune more aggressively. For a detailed description of the method we used to determine $x$, please refer to Appendix D.1.

### 4.3 PRESERVING THE MUTUAL INFORMATION BETWEEN ADJACENT LAYERS IN PRACTICE

In this Section, we apply the methods discussed above and describe how to use TERC to preserve MI between a pair of adjacent layers. As discussed, TERC with MI ordering dictates that, to remove a node, the following should be satisfied: $I(\mathcal{X}_{L-1} \backslash X_{L-1}^i; \mathcal{X}_L) = I(\mathcal{X}_{L-1}; \mathcal{X}_L)$. In Section 4.2.2, we describe the method we use to estimate MI. By combining these representations, we can update the condition we wish to approximate:

$$I(\mathcal{X}_l; \mathcal{X}_{l+1}) = I(\mathcal{X}_l \backslash X_l^i; \mathcal{X}_{l+1}) \quad \text{(original condition as in TERC)},$$

$$\mathbb{E}[l(f(\emptyset), \mathcal{X}_{l+1})] - \mathbb{E}[l(g(\mathcal{X}_l), \mathcal{X}_{l+1})] = \mathbb{E}[l(f(\emptyset), \mathcal{X}_{l+1})] - \mathbb{E}[l(h(\mathcal{X}_l \setminus X_l^i), \mathcal{X}_{l+1})], \quad (1)$$

$$\mathbb{E}[l(g(\mathcal{X}_l), \mathcal{X}_{l+1})] = \mathbb{E}[l(h(\mathcal{X}_l \setminus X_l^i), \mathcal{X}_{l+1})] \quad \text{(estimated condition)}.$$

Equation 1 demonstrates the simplification possible when $I(X;Y) \approx \mathbb{E}[l(f(\emptyset), Y)] - \mathbb{E}[l(f(X), Y)]$ is substituted into $I(\mathcal{X}_l; \mathcal{X}_{l+1}) = I(\mathcal{X}_l \backslash X_l^i; \mathcal{X}_{l+1})$. Our condition becomes a simple comparison of a loss function with and without a node. To calculate the updated function h and evaluate the loss l, we use a simple MLP.

Using this updated condition, we apply TERC with MI ordering, which can be described as follows: initially, we order the nodes in descending order of the loss achieved when using just this variable to predict the downstream layer. Then, we train an MLP to reconstruct the activations of the downstream layer from the entirety of the upstream layer's activations. Like Gadgil et al. (2024), we sequentially mask individual upstream nodes and re-train this MLP (although, not to the same extent as in the first instance) to determine whether the loss function drops back below its original value. If it fails to recover, this implies that, without the activations of this node, we are unable to reconstruct the activations of the downstream layer. In this case, the variable is considered informative and should be retained in the network and in the set $\mathcal{X}_l'$. Otherwise, the node is removed.

In the introduction, we outlined the challenges of ranking neurons. Such methods overlook the impact that removing a node has on the importance of those remaining, while also causing layer collapse. MIPP, overcomes these two problems respectively due to the following mechanistic features. Firstly, MIPP performs *per-node function discovery*. Some new function (labeled h in Equation 1) is discovered for each node removed, implying a non-static ranking, where the removal of all previous nodes is considered when evaluating whether to remove future nodes. Secondly, MIPP also exploits *adjacent layer dependence*. MIPP only removes nodes that are not essential for reconstructing the next layer. As more nodes are pruned, those remaining become increasingly vital in the reconstruction, preventing layer collapse.

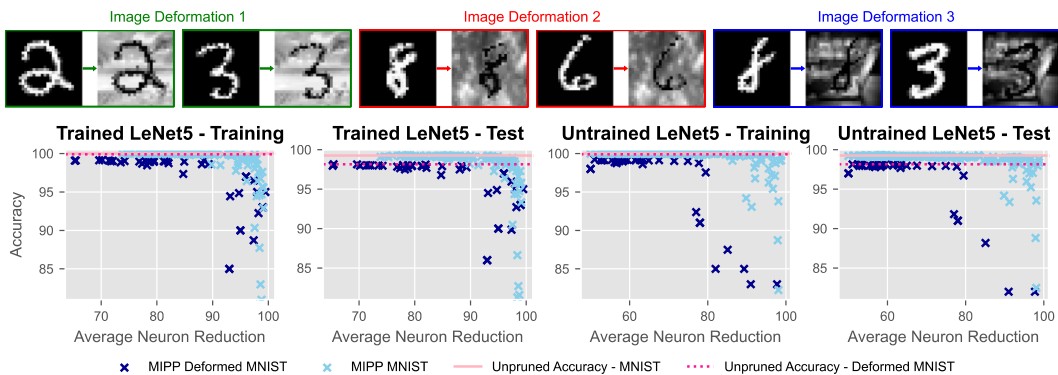

Figure 1: *Top.* Deforming MNIST for increased image complexity. These transformations were applied randomly with equal probability and then kept consistent during training, pruning, and re-training. *Bottom.* Changes in pruning ability of MIPP caused by image deformation.

### 4.4 PRESERVING THE MUTUAL INFORMATION FROM OUTPUTS TO INPUTS

Thus far we have explicitly described how we use TERC with MI ordering and the estimation techniques described in Section 4.2.2, to preserve MI between the activations of adjacent layers. This process is repeated for each pair of layers. However, to prune the entire model, by preserving the MI between pairs of layers, one could start from the input layer and move to the output layer or vice versa. In this section, like Luo et al. (2017), we argue for the second option, providing both theoretical and practical arguments.

**Theoretical argument.** In a NN, because each layer is a direct function of its predecessor, these pairs share perfect MI. In this case $I(\mathcal{X}_l; \mathcal{X}_{l+1}) = H(\mathcal{X}_{l+1})$ (Cover, 1999). Therefore, the networks layers can only reduce in entropy from inputs to outputs (Tishby & Zaslavsky, 2015; Shwartz-Ziv & Tishby, 2017). Suppose we take the first approach, pruning from inputs to outputs. Our goal is to prune the first layer ($\mathcal{X}_1$), such that the result can be used to reconstruct the activations of the second layer ($\mathcal{X}_2$). Since the second layer has not yet been pruned, it may retain superfluous information, which is then maintained in the activations of the first layer during pruning. In contrast, if we take the second approach, we begin by pruning the activations in layer $\mathcal{X}_{L-1}$. The information in $\mathcal{X}'_{L-1}$ (its pruned version) has been preserved due its ability to reconstruct exclusively the outputs. Upon moving onto the next pair, we prune layer $\mathcal{X}_{L-2}$ based on the entropy in the layer $\mathcal{X}'_{L-1}$. Notably though, this has already been reduced by the first pairwise pruning step. By this recursive logic, it is clear how even when pruning the first layer, we are still only preserving the entropy required to reproduce the outputs, and only the outputs.

**Practical argument.** We now present the more practical reason to prune from outputs to inputs rather than vice versa. Under this scheme we aim to evaluate the condition $I(\mathcal{X}'_l; \mathcal{X}'_{l+1}) = I(\mathcal{X}_l; \mathcal{X}'_{l+1})$, rather than $I(\mathcal{X}'_l; \mathcal{X}_{l+1}) = I(\mathcal{X}_l; \mathcal{X}_{l+1})$ which would be appropriate forward pruning was conducted. In the former case, we apply our MLP to predict a layer whose dimensionality has already been reduced. This increases efficiency by mitigating the effects of the curse of dimensionality (Bellman & Kalaba, 1959). We have now presented the steps used to explain MIPP. In Algorithm 1, we synthesize this information more formally. Notably, the utility of MIPP can extend beyond just pruning. By verifying which pixels transfer entropy to the activations of the pruned first layer, MIPP also possesses the ability select features. We present the corresponding experiments in Appendix E.1.

## 5 EVALUATION

**Models, datasets and baselines.** CNNs are characterized by multivariate filters in addition to univariate nodes. In Appendix E.2 we discuss how our method can be adapted so that it preserves information between filters. We begin by applying our method to the simple LeNet5 architecture detecting variations of the MNIST dataset (LeCun et al., 1998). We then assess its ability to prune VGG11, ResNet18 and ResNet34 networks trained on the CIFAR10 dataset (He et al., 2016). We

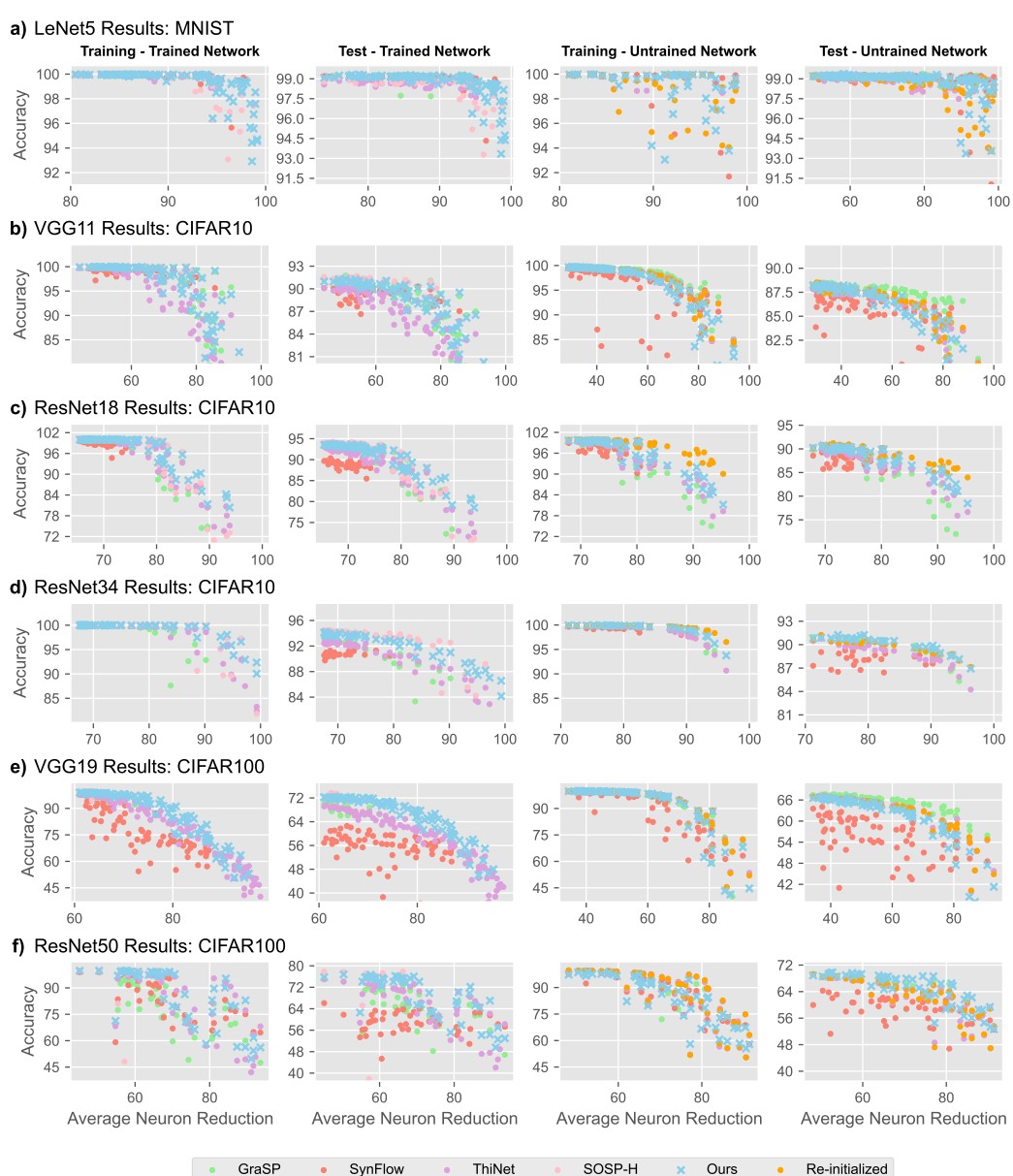

Figure 2: Pruning results for ours and other methods as applied to multiple datasets and models.

then evaluate more complex models, specifically ResNet50 and VGG19 on the CIFAR100 dataset (Krizhevsky, 2009; Simonyan & Zisserman, 2015). Finally, we examine our method's effectiveness in pruning a pre-trained ResNet50 model on the ImageNet dataset (Deng et al., 2009). For models trained on datasets smaller than ImageNet, we compare the performance of our method to SynFlow (Tanaka et al., 2020), GraSP (Wang et al., 2022), ThiNet (Luo et al., 2017) and SOSP-H (Nonnenmacher et al., 2022), due to memory limitations we only compare to ThiNet on larger datasets. SOSP-H was not designed for untrained networks and so, for these experiments, we instead use a re-initialized baseline. Both GraSP and SynFlow are unstructured; in order to make them structured, we apply L1-normalization to all the weights associated with a node. MIPP selects nodes based on whether their activations transfer entropy to those of the subsequent layer. This approach inherently establishes a unique PR for each run, which we adopt as the global PR for our baseline methods. ThiNet cannot determine layer-wise PR; therefore, we apply a uniform PR across all layers.

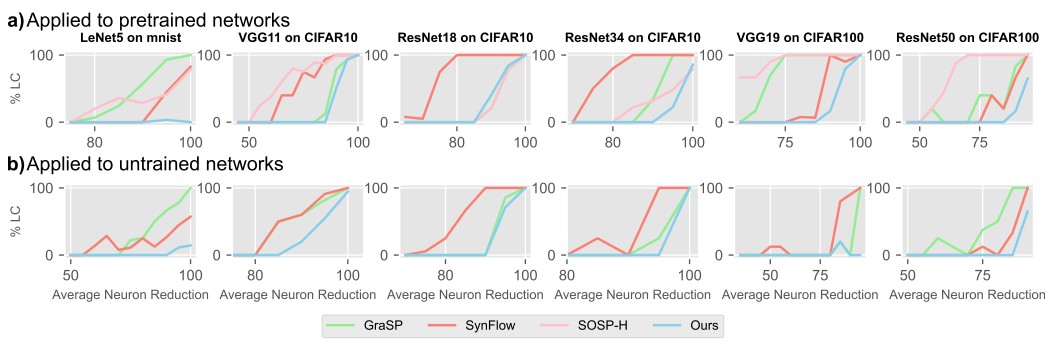

Figure 3: The percentage of runs that led to untrainable layer collapse. Specifically, we bin runs by the percentage of neurons removed, where one bin contains all the runs within a 5% increment. We then calculate the percentage of these runs that lead to layer collapse.

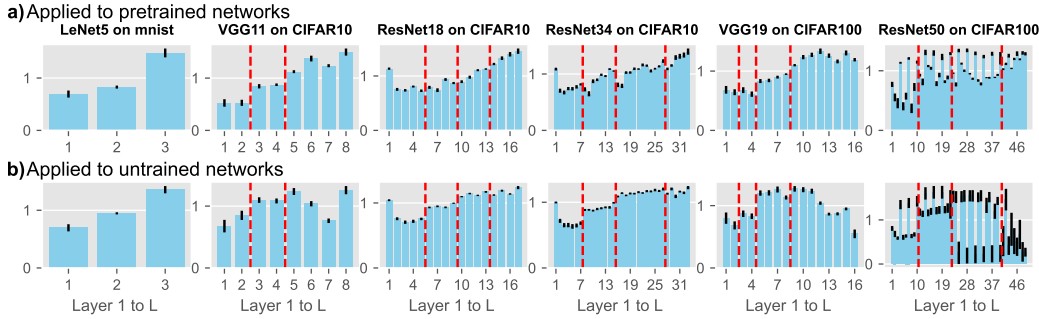

Figure 4: These experiments demonstrate the per-layer PR selected by MIPP. For the different layer-wise PRs we divide them by the average of all the layers in order to normalize. We omit results on ImageNet for space and clarity.

**LeNet5 on MNIST.** We evaluate our method's ability to prune a LeNet5 architecture trained on MNIST, and an untrained LeNet5 with MNIST acting as inputs. For both the trained and untrained networks, as shown in Figure 2 a), we observe that MIPP consistently selects nodes and filters that lead to competitive results. In Figure 3, we demonstrate that MIPP is the method most robust to layer collapse, producing trainable models even at sparsity levels above 95 %.

**LeNet5 on deformed MNIST.** MIPP effectively preserves and compresses the information encoded in network activations. In untrained networks, these activations solely reflect the information present in the input data. If these inputs are characterized by information relevant to the classification task, MIPP remains applicable. For instance, in the MNIST dataset, the informative pixels assist the classification task, while the remaining pixels, on the outskirts of the image, are constantly black and contain no information. In such cases, our method selectively preserves the neurons whose activations correspond to informative pixels. On the other hand, the converse is also true; our method is inapplicable to models whose input data contains information not relevant for the classification task. Consequently, if the input data is complex, MIPP's ability to prune at initialization is reduced. To demonstrate this effect, in Figure 1 we present experiments that investigate the effects of deforming MNIST. In alignment with our hypothesis, we observe a reduction in our ability to prune an untrained network but not a trained network. When MIPP is applied to trained networks, it can successfully prune to high sparsity levels, regardless of whether the dataset has been deformed. The same is not true for untrained models, where we observe an early drop in the deformed dataset classification accuracy.

**VGG11 on CIFAR10.** We now investigate our method's ability to prune a VGG11 trained on CIFAR10. These results are presented in the left-most two graphs of Figure 2 b). We observe that MIPP leads to a better performing model at train-time, and test time.

Moreover, MIPP is more resistant to layer collapse effects in untrained networks. In Figure 3, even at a sparsity level above 90 %, untrainable bottlenecks remain rare. For the untrained network MIPP remains competitive but is slightly out-performed by both GraSP and reinitialize baselines.

**ResNet18 on CIFAR10.** In Figure 2 c), we provide a comparison of the pruning performance between MIPP and the baseline methods on a ResNet18 model trained with CIFAR10. We observe that our method outperforms the baselines when applied to pre-trained networks and is competitive for newly initialized models. As illustrated in Figure 3, MIPP only causes layer collapse at sparsity levels much higher than competing techniques. This occurs due to MIPP's adjacent layer objective.

**ResNet34 on CIFAR10.** For this example we again observe the advantages of using our method, particularly at high sparsity levels. Nonetheless, SOSP-H does outperform MIPP at test time if pruning at lower sparsity levels - between 80-90%. SOSP-H's generalizability is due to its ability to establish performant layer-wise PRs, aggressively pruning the later layers. However, at ultra-high sparsity levels, these same layers collapse, causing the results in Figure 3. In Figure 4 we observe block-based PRs. This is particularly apparent for the untrained model. However, in this case there is also the presence of intra-block PR patterns: in the last three blocks, layers alternate between more and less pruned. This occurs due to the effect of the skip connections in a residual network, acting to stabilize the activations and increasing the PR. In Figure 5 we provide a pictorial explanation of the ResNet structure, from which it is possible to understand why this intra-block structure has a periodicity of two.

**VGG19 on CIFAR100.** In the two left-most graphs of Figure 2 b) it can be observed that MIPP outperforms the baselines. As discussed, increasing the complexity of the dataset decreases the ability to prune untrained models using MIPP. For these reasons, GraSP (designed to be used at initialization) and re-initialization marginally outperform MIPP at high sparsity levels on untrained networks.

**ResNet50 on CIFAR100 and ImageNet.** In Figure 2 f) we observe that, despite noisy results, MIPP generally outperforms baselines, particularly on untrained networks. In Figure 4, we observe intra-block pruning patterns. This is a simple consequence of the ResNet50 structure, presented in Figure 5. Specifically, one in every three layers is pruned more aggressively as one in every three layers is more overparameterized. From the results on ImageNet in Figure 6, it is clear that we are able to prune even on large datasets and models. MIPP generally outperforms ThiNet at test time due to its ability to establish layer-wise PRs. This is because CNNs are known to generalize better when their remaining nodes are concentrated in the early layers. Overall, these experimental results demonstrate the ability of MIPP to surpass state-of-the-art performance when pruning trained NNs and to establish layer-wise PRs that encourage generalizability, as evidenced in Figure 2.

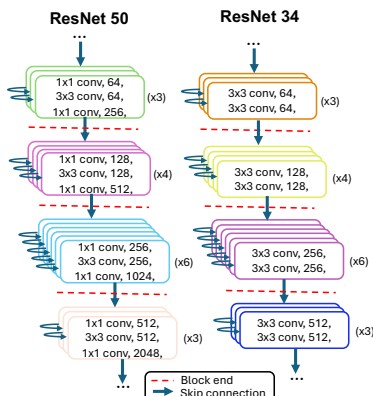

Figure 5: ResNet34 and ResNet50 structures, explaining the periodicity of the per-layer PRs established using our method.

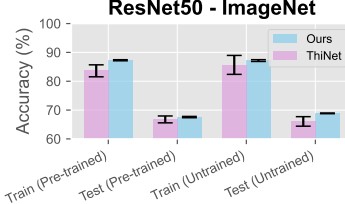

Figure 6: Performance evaluation on ImageNet, with an average PR of $71.1 \pm 0.81\%$ and $55.6 \pm 0.62$ on the pre-trained and not pre-trained networks respectively.

# 6 CONCLUSION

Current node selection methods rank nodes before selecting the top-$k$. These static ranking systems not only fail to consider the effect of removing nodes on the current potential ranking but also often lead to layer collapse, motivating the need for a more dynamic node selection method. Consequently, we have introduced MIPP, an activation-based pruning method that removes neurons or filters from layers if they fail to transfer entropy to the subsequent layer. Consequently, MIPP preserves MI between the activations of adjacent layers. We have applied the proposed method to a variety of datasets and models. Our experimental evaluation has demonstrated the effectiveness of MIPP in pruning trained and untrained models characterized by differing complexities.

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

## A  NOTATION

Table 1: Summary of Notational Conventions

| Type | Notation |
|---|---|
| Vectors | $\boldsymbol{x}^n$ |
| Matrices | $\boldsymbol{X}^{m \times n}$ |
| Random Variables | $X$ |
| Instances of Random Variables | $x$ |
| Sets of Jointly Sampled Random Variables | $\mathcal{X}$ |
| Functions | x |
| Nested Functions | X |

## B  ALGORITHMS

In this section, we present not only the overall MIPP algorithm but also TERC with MI ordering algorithm, which maintains the MI between adjacent layers in a network.

---

**Algorithm 1** MIPP.

**Input**: Activations of all layers: $\mathcal{X}_l$.  **Output**: $\mathcal{M}$ (a desirable set of node masks).

1: Initialize empty set of masks: $\mathcal{M} = \emptyset$.
2: **for** $l \in [L-1, 0]$ **do**
3:   $\mathcal{X}_l' = $ Algorithm 2$(\mathcal{X}_l, \mathcal{X}_{l+1})$
4:   **for** $i \in [0, I]$ **do**
5:     $\boldsymbol{m}_l^n(i) = \begin{cases} 0 & \text{if } X_l^i = 0, \\ 1 & \text{otherwise.} \end{cases}$
6:   **end for**
7:   $\mathcal{M} = \mathcal{M} \cup \boldsymbol{m}_l^n$
8: **end for**
9: **return** $\mathcal{M}$

---

**Algorithm 2** TERC with MI ordering.

**Input**: Activations of layers $L$ and $L-1$: $\mathcal{X}_L$ and $\mathcal{X}_{L-1}$. **Output**: $\mathcal{X}_{L-1}'$ (a desirable subset of nodes).

1: Initialize $\mathcal{X}_{L-1}' = \text{sort}_{\text{desc}}\left(\mathcal{X}_{L-1}, I(X_{L-1}^i; \mathcal{X}_L)\right)$
2: **for** $X_{L-1}^i \in \mathcal{X}_{L-1}$ **do**
3:   **if** $I(\mathcal{X}_{L-1}' \backslash X_{L-1}^i; \mathcal{X}_L) = I(\mathcal{X}_{L-1}; \mathcal{X}_L)$ **then**
4:     $\mathcal{X}_{L-1}' = \mathcal{X}_{L-1}' \backslash X_{L-1}^i$
5:   **end if**
6: **end for**
7: **return** $\mathcal{X}_{L-1}'$

---

## C  PROOF OF THEOREM 1

In this section we prove Theorem 1. To begin, we remind the reader that we aim to preserve the MI between layers such that:

$$I(\mathcal{X}_{L-1}'; \mathcal{X}_L) = I(\mathcal{X}_{L-1}; \mathcal{X}_L), \tag{2}$$

which, given the relationship $I(X; Y) = \sup_f \left( \mathbb{E}[f(X \mid Y) - \log \mathbb{E}[e^{f(X)}]) \right)$, becomes:

$$\sup_g \left( \mathbb{E}[g(\mathcal{X}_{L-1}) \mid \mathcal{X}_L] - \log \mathbb{E}[e^{g(\mathcal{X}_{L-1})}] \right) = \sup_f \left( \mathbb{E}[f(\mathcal{X}'_{L-1}) \mid \mathcal{X}_L] - \log \mathbb{E}[e^{f(\mathcal{X}'_{L-1})}] \right). \quad (3)$$

However, we know that there exists a function $g$ such that $g(\mathcal{X}_{L-1}) = \mathcal{X}_L$. Therefore, we can rewrite the above such that:

$$\left( \mathbb{E}[\mathcal{X}_L \mid \mathcal{X}_L] - \log \mathbb{E}[e^{\mathcal{X}_L}] \right) = \sup_f \left( \mathbb{E}[f(\mathcal{X}'_{L-1}) \mid \mathcal{X}_L] - \log \mathbb{E}[e^{f(\mathcal{X}'_{L-1})}] \right),$$

$$\mathcal{X}_L - \log \mathbb{E}[e^{\mathcal{X}_L}] = \sup_f \left( \mathbb{E}[f(\mathcal{X}'_{L-1}) \mid \mathcal{X}_L] - \log \mathbb{E}[e^{f(\mathcal{X}'_{L-1})}] \right). \tag{4}$$

The only circumstances under which Equation 4 holds is if $f(\mathcal{X}'_{L-1}) = \mathcal{X}_L$, thereby proving Theorem 1.

## D  FURTHER EXPERIMENTAL SETTINGS

### D.1  SELECTION OF THE PRUNING RATIO

MIPP selects a node if it transfers entropy to the subsequent layer. This prevents us from defining a pruning ratio and selecting the top-$k$ variables in a global fashion. Despite this, for our study we still affected the pruning rate by changing the confidence required to remove a node ($x\%$). For example, if $x\% = 99\%$ one must be very confident that entropy is transferred to the following layer to maintain a node in the NN, promoting an aggressive pruning strategy. In this section, we explain how we derived values of $x\%$.

Specifically, we had 20 different values of $x\%$ for each set of experiments. The lowest possible value being $x\% = 50\%$, where we would then generate 20 different values of $x_n\%$ (where $n \in [0, 20]$) using the following equation: $x_n = x_0 + \sum_{i=1}^{n}(1 - x_{i-1}) \cdot r$, where $r = 0.5$ and $x_0 = 50$. Using this method we generate 20 values of $x$ that approach $100\%$ confidence at a decreasing rate.

### D.2  DATA AUGMENTATION TECHNIQUES

For the CIFAR-10 dataset, we applied standard data augmentation techniques, which included random cropping with padding and random horizontal flipping. These augmentations are commonly used to enhance model generalization by introducing variations in the training data. In the case of CIFAR-100, we employed additional augmentation methods beyond the standard techniques. Specifically, we used mixup (Zhang et al., 2018), which creates virtual training examples by combining pairs of images and their labels, and cutout (DeVries & Taylor, 2017), which randomly masks out square regions of an image to simulate occlusion and encourage the network to focus on more distributed features. These advanced techniques were included to further enhance performance due to the increased complexity of the CIFAR-100 dataset.

### D.3  HYPERPARAMETERS

#### D.3.1  VISION TRAINING AND RE-TRAINING

Please refer to Table 2.

Table 2: Comparison of training parameters across datasets.

| Dataset | MNIST | CIFAR10/100 | ImageNet |
|---|---|---|---|
| Solver | SGD (0.9, 1e-4) | SGD (0.9, 5e-4) | SGD (0.9, 1e-4) |
| Batches | 100 | CIFAR10: 128, others: 256 | 1024 |
| LR | 1e-2, [30,60], #epochs:90 | 1e-1, [100,150], #epochs:200 | 1e-1, [30,60], #epochs:90 |
| LR (re-train) | 1e-2, [30,60], #epochs:90 | 1e-2, [60,90], #epochs:120 | 1e-2, [30,60], #epochs:75 |

### D.3.2 PRUNING

As explained in the main paper, our method consists of using an MLP to predict the activations of a layer based on its predecessor. We then mask the features and re-train the MLP to see if the loss will drop back below the original. We therefore require the number of iterations and the size of the MLP fit initially and the number of iterations required for re-training. For all layers and models we fit three MLPs with two hidden layers with 256 nodes. For the initial training step we used 1500 iterations. For the re-training steps once the mask has been applied we use 20 iterations for VGG based models and 150 iterations for others. At the start of our algorithm, we also rank the features based of their MI. For this calculation we again use the same MLP structure but only for 35 iterations due to time constraints. We use our method to prune all linear and convolutional layers. We prune the batch-normalization nodes associated with nodes in linear/convolutional layers, while skip connections in ResNets remain unaffected.

## E  FURTHER EXPERIMENTS

### E.1  FEATURE SELECTION EXPERIMENTS

In this Section, we investigate MIPP's ability to select features, specifically reviewing the pixels it identifies from the MNIST dataset. MIPP selects features in the exact same manner it selects nodes, by verifying whether entropy is transferred from these variables to those of the subsequent layer. In this case, the subsequent layer is the first layer in the network. In Figure 7, we observe what appears as a significantly stochastic pixel selection. However, a tendency to select features from the right hand side of the image can be observed. This aligns with experiments done by Covert et al. (2020), which revealed there are more pixels correlated with the dataset labels in this area. Despite this signal, it is non-intuitive to observe pixels in the right-most column being selected while those in the center of the image are not. In spite of this, we observe good performance, MIPP is able to achieve state-of-the art test accuracy in the scenarios in which 90% and 75% of neurons and pixels have been removed, respectively. In Figure 8, we present the average accuracy achieved when we prune models using MIPP and our baselines. Meanwhile, in Figure 9 we present the layer collapse rates. Unlike the main body of the paper, in both of these figures we have also used MIPP to select pixels. In both figures we observe that MIPP outperforms the baselines. This is because, unlike any of the baselines, the features are selected in a manner that is dependent on the pruned model. MIPP can compress both features and the underlying model simultaneously such that the results are compatible, preventing ML practitioners from having to use different methods for feature and model compression. Often, combining compressed input and compressed models can lead to performance degradation. SOSP-H is often close to state-of-the-art when pixels have not been selected; however, in these experiments it performs poorly. This is because the gradients calculated with respect to the input are more sparse.

### E.2  COMPARING FEATURE COLLAPSE FUNCTIONS

Vision models, being vastly overparameterized, have become of key importance when evaluating pruning algorithms (Real et al., 2019; Wang & Fu, 2023; Wang et al., 2023). Unlike the nodes of MLPs, filters in CNNs are multivariate. To preserve the MI between layers, we have two options: we can either flatten the square filters into an array of random variables, which retains all possible information but is computationally expensive, or collapse each filter into a single value using a function. Given the complexity of the former approach, we opt for the latter, showing that, despite the potential loss of information, MIPP remains highly effective. To find the function that preserves the most information, we compare L1, L2, mean and std functions, for the exact form of these functions please refer to Table 4 (Molchanov et al., 2017; Liu et al., 2017).

If Figure 10, it can be observed that for whichever function is chosen, MIPP remains performant. However, the L1-normalization function consistently demonstrates an ability to prune to high sparsities while layer collapse remains rare. For this reason, we adopted this function throughout when collapsing convolutions.

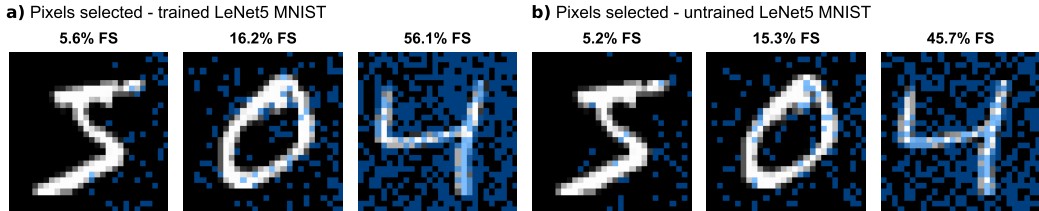

Figure 7: Visual representation of the features selected using MIPP at different sparsities (blue implies selected).

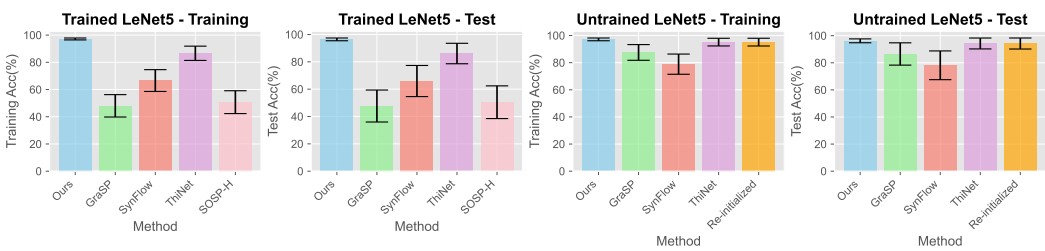

Figure 8: Average accuracy at train and test time using each method when features have been selected.

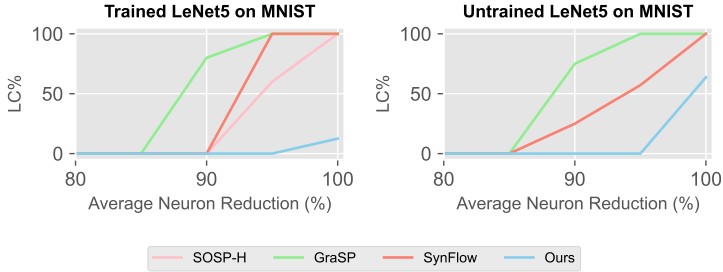

Figure 9: % of runs that lead to layer collapse when features have been selected.

### E.3 Layer Wise Pruning Ratios Established Using Other Methods

In the main paper, we present the per-layer PRs MIPP obtains. In Figure 11, we present these results for the other methods taken into consideration.

## F Comparison of Baseline Characteristics

In an attempt to help the reader better understand the relationships between MIPP and the existing literature, we present Table 4, which compares different characteristics of our method with the chosen baselines.

Table 3: Summary of Collapse Methods. $x_l^i$ can be interpreted as one realization of the random variable $X_l^i$, while $\mathbf{O}_l^{h \times w, i}$ is a matrix of filter activations associated with the same filter or node as $x_l^i$.

| Collapse Method | Mathematical Notation |
|---|---|
| Mean | $x_l^i = \dfrac{1}{HW} \sum\limits_{h=1}^{H} \sum\limits_{w=1}^{W} \left| \mathbf{O}_l^{h \times w, i} \right|$ |
| Standard Deviation | $x_l^i = \sqrt{\dfrac{1}{HW} \sum\limits_{h=1}^{H} \sum\limits_{w=1}^{W} \left( \left| \mathbf{O}_l^{h \times w, i} \right| - \mu_{h,w} \right)^2}$ |
| L2-Norm | $x_l^i = \sqrt{\sum\limits_{h=1}^{H} \sum\limits_{w=1}^{W} \left( \mathbf{O}_l^{h \times w, i} \right)^2}$ |
| L1-Norm | $x_l^i = \sum\limits_{h=1}^{H} \sum\limits_{w=1}^{W} \left| \mathbf{O}_l^{h \times w, i} \right|$ |

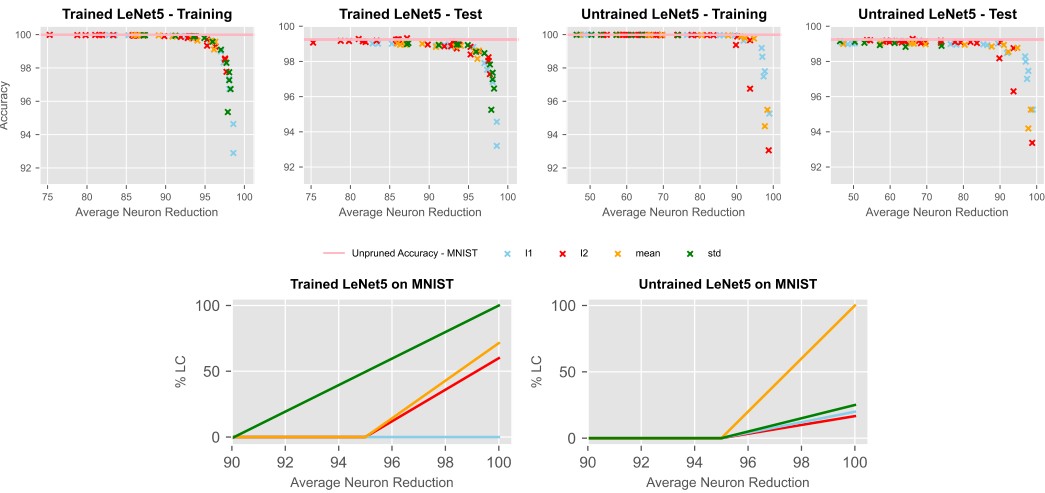

Figure 10: *Top.* Comparing functions used to collapse filter activations on their ability to promote succesful pruning. *Bottom.* Comparing functions used to collapse filter activations on their ability to avoid layer collapse.

Table 4: Comparison of Pruning Methods.

| Feature | MIPP (Ours) | ThiNet | SOSP-H | GraSP | SynFlow |
|---|---|---|---|---|---|
| Activation-Based | ✓ | ✓ | ✗ | ✗ | ✗ |
| Adjacent Layer-Based | ✓ | ✓ | ✗ | ✗ | ✗ |
| Structured | ✓ | ✓ | ✓ | ✗ | ✗ |
| Able to establish layer-wise PRs | ✓ | ✗ | ✓ | ✓ | ✓ |
| Layer Collapse Resistant | ✓ | ✗ | ✓ | ✓ | ✓ |
| Data Dependent | ✓ | ✓ | ✓ | ✓ | ✗ |
| At Initialization | ✓ | ✗ | ✓ | ✓ | ✓ |

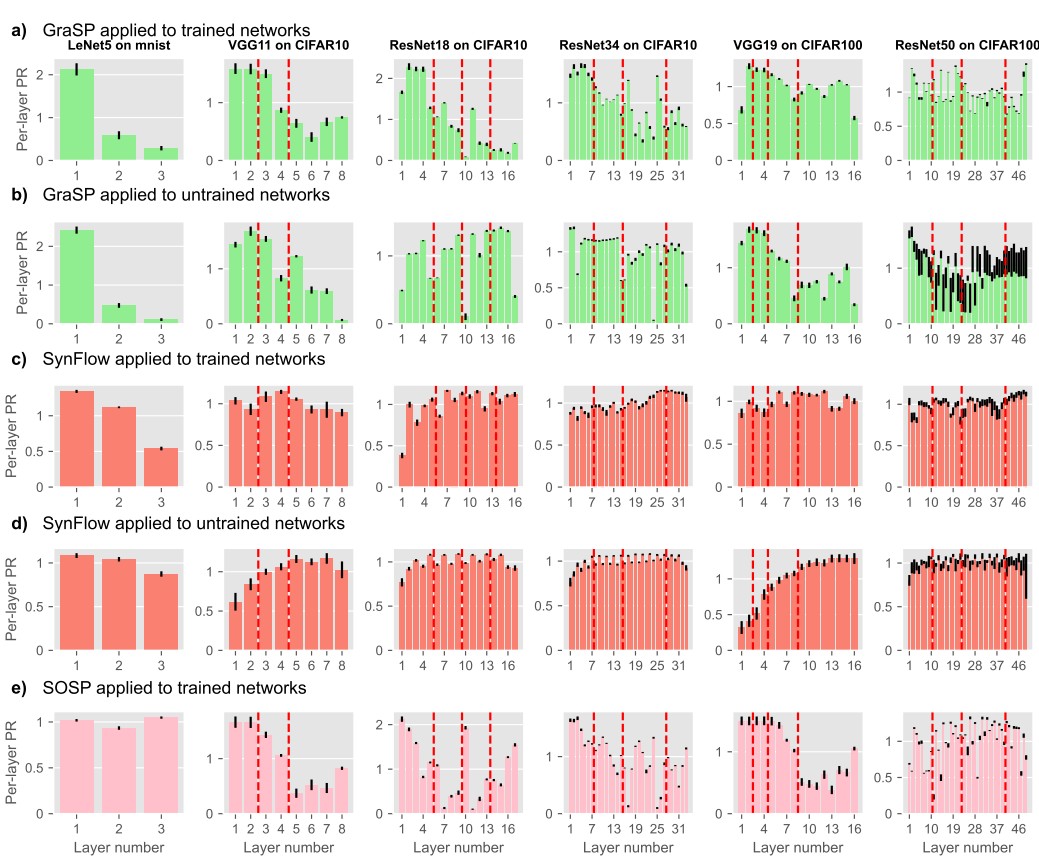

Figure 11: Layer-wise pruning ratios. Normalized by division of the average PR achieved for that run.

