# OpenReview forum: "Mutual Information Preserving Neural Network Pruning"
_ICLR.cc/2025/Conference — Submitted to ICLR 2025_

### Official Review · Reviewer_ZnsM · 2024-11-01

**Soundness:** 3
**Presentation:** 3
**Contribution:** 2
**Rating:** 3
**Confidence:** 4

**Summary:**

This paper introduces MIPP (Mutual Information Preserving Pruning), a structured pruning method for neural networks. The key idea is to preserve mutual information between adjacent layers' activations during pruning by selecting nodes that transfer entropy to the subsequent layer.
The method operates by iteratively pruning from outputs to inputs, using transfer entropy redundancy criterion (TERC) with MI ordering to select nodes. Comprehensive experiments validate MIPP's effectiveness on both trained and untrained networks.

**Strengths:**

1. The overall writing is clear for effective visualizations and well-structured presentation.
2. The paper conducts a wide range of experiments to validate the algorithm.

**Weaknesses:**

1. Some important references [a] are missing, which makes the novelty of the paper questionable. For example, [a] is also about using the mutual information to do filter pruning, what is the difference? Whether the proposed paper achieve higher performance? Why and How?

2. The compared method is rather old. The authors claim "For models trained on datasets smaller than ImageNet, we compare the performance of our method to SynFlow (Tanaka et al., 2020), GraSP (Wang et al., 2022), ThiNet (Luo et al., 2017) and SOSP-H (Non- nenmacher et al., 2022),". Why not include paper published in 2024 [b].

3. Performance is not good. In [a], the performance on ResNet-50 on ImageNet is much higher than Thinet. Why just compare Thinet in Figure 6?

[a] Enhancing CNN efficiency through mutual information-based filter pruning, Digital Signal Processing 2024
[b] Auto-Train-Once: Controller Network Guided Automatic Network Pruning from Scratch

**Questions:**

See weaknesses

---

> ### Author Response · Authors · 2024-11-22
> **Response**
>
> ## Responses to Reviewer Comments
> ### Reviewer Comment 1:
> “Some important references [a] are missing, which makes the novelty of the paper questionable. For example, [a] is also about using the mutual information to do filter pruning, what is the difference? Whether the proposed paper achieve higher performance? Why and How?”
>
> ### Response
> The suggested paper [a] aims to select nodes based on their conditional mutual information (MI) with the target and a previously selected node. This places the paper in the category described in Section 2: “Activation-based pruning methods commonly view the activations as features and the outputs as targets, before ranking and selecting the top-k nodes in a global or local manner.” Our method, however, estimates the MI between activations in adjacent layers, as this function requires fewer parameters to approximate. Furthermore, the method in [a] conditions the information on a single selected node. Finally, [b] was published in June 2024 (at CVPR). According to the ICLR guidelines, this qualifies as “contemporaneous” research (published within 4 months).
>
>
> ### Reviewer Comment 2:
> “Performance is not good. In [a], the performance on ResNet-50 on ImageNet is much higher than Thinet. Why just compare Thinet in Figure 6?”
>
> ### Response
> We only compared to one baseline on ImageNet due to the large amount of GPU time required to train a model on ImageNet (even after pruning) given our computational budget at our institution.

---

> > ### Comment · Reviewer_ZnsM · 2024-11-24
> >
> > Thanks for your response.
> >
> > After reading the response and other reviewers' reviews, I'd like to keep my score since the response didn't provide the required comparison and experiments.

---

### Official Review · Reviewer_NS2s · 2024-11-03

**Soundness:** 2
**Presentation:** 2
**Contribution:** 2
**Rating:** 3
**Confidence:** 4

**Summary:**

The paper introduces Mutual Information Preserving Pruning (MIPP), an activation-based pruning method that maintains mutual information between adjacent layers in neural networks, ensuring retrainability. Unlike traditional methods, MIPP dynamically selects nodes based on their contribution to information transfer, addressing limitations such as layer collapse and lack of adaptability. Experimental results demonstrate that MIPP outperforms state-of-the-art pruning techniques on various vision models, including ResNet50 on ImageNet, with implementation details to be released upon publication.

**Strengths:**

This paper provides an interesting perspective on neural network pruning. It considers the activations of the downstream layers, which allows pruning on trained and untrained networks; the idea is interesting.

**Weaknesses:**

1. Authors claim the method is compared with state-of-the-art techniques, yet most literature is from before 2022; many recent works, such as PHEW or NPB in Pruning at Initialization, and indeed, there are more works on pruning on trained networks in recent years. I strongly suggest that authors provide more valid reviews of recent works.
2. Although the method is interesting because it works for both trained and untrained networks, the motivation for this is not clear.
For PaI, tasks aim to find networks before training to reduce training costs, while post-trained pruning aims to preserve the best performance on trained networks.  Is MIPP better than SoTA methods on each side?

**Questions:**

1. Can the author compare MIPP in PaI and post-trained pruning in separate settings with SoTA methods in recent years, such as 2023 or 2024?
2. Most experiments are performed on small datasets and networks; showing the pruning task on more computationally intensive structures or datasets, such as the Efficentnet B7 and Imagenet1K tasks, would be better. As the author claimed, the methods work for trained networks; pruning on a network that is pre-trained in a large-scale dataset such as Imagenet21K could be interesting.
3. Will this method work in Vision transformers?

---

> ### Author Response · Authors · 2024-11-22
> **Response**
>
> ## Responses to Reviewer Comments
>
> ### Reviewer Comment 1:
> “Authors claim the method is compared with state-of-the-art techniques, yet most literature is from before 2022; many recent works, such as PHEW or NPB in Pruning at Initialization, and indeed, there are more works on pruning on trained networks in recent years. I strongly suggest that authors provide more valid reviews of recent works.”
>
> ### Response
> We thank the reviewer for the comment. We did not select PHEW because, although it is an activation-based structured pruning method (aligned with MIPP), it is designed to be applied without training data, which is not the case for MIPP. Therefore, we considered ThiNet the most applicable baseline, as it is not only structured, activation-based, and widely adopted, but also designed to be data-dependent. The PaI paper appears relevant and will help inform our paper revisions.
>
> The reviewer says that our baseline are old but in fact they were published in 2017, 2020 an 2022 (two of them).
>
>
> ### Reviewer Comment 2:
> “Although the method is interesting because it works for both trained and untrained networks, the motivation for this is not clear. For PaI, tasks aim to find networks before training to reduce training costs, while post-trained pruning aims to preserve the best performance on trained networks. Is MIPP better than SoTA methods on each side?”
>
> ### Response
> We tried to make this clear in the results section for MNIST; however, we will emphasize the following points further in a revised version of the paper. MIPP is always applicable and performant in the context of trained networks, while this is not always the case with untrained networks.
>
> MIPP maintains the information flow between the activations of adjacent layers in a network. It prunes uninformative nodes, whose activations may be irrelevant or redundant. In a trained network, the activations are optimized to complete the task at hand. Therefore, MIPP effectively preserves useful information, leading to competitive performance.
>
> On the other hand, this is not always the case when applied to untrained networks. If untrained, the information in the network reflects the information in the input images. Unlike in the case of the trained network, the information in these input images may not necessarily be useful for the classification task. In this case, MIPP may preserve information that will not contribute to the training process, impeding its performance.
>
> Hence, MIPP can be applied to both trained and untrained networks; however, as dataset complexity increases, it becomes less applicable to untrained networks. The reason we still see good results in some cases for untrained networks is due to MIPP’s ability to achieve competitive layer-wise pruning ratios. We tried to emphasize this with the experiments presented in Figure 1.

---

> > ### Comment · Reviewer_NS2s · 2024-11-25
> >
> > Thank you for your response. We appreciate your acknowledgement of the novelty and interest in the MIPP concept. We agree that the paper would benefit from additional experiments to validate its performance. Specifically, comparing MIPP to previous methods—such as pruning at initialization and post-training pruning—would be valuable in highlighting its strengths in terms of efficiency and performance compared to existing approaches. I will keep my score and encourage the author to continue improving the current paper for future venues.

---

### Official Review · Reviewer_7FoL · 2024-11-03

**Soundness:** 2
**Presentation:** 2
**Contribution:** 2
**Rating:** 5
**Confidence:** 3

**Summary:**

This paper proposes a pruning approach based on mutual information between the activations of adjacent layers. The proposed approach has been evaluated on a number of models and datasets.

**Strengths:**

+ The motivation of the paper is clear - global pruning approaches do have their limitations and the proposed approach can effectively avoid those.
+ The idea of looking at the MI between activations of adjacent layers is interesting. It also makes sense to consider nodes that can maintain such MI.
+ The theoretical analysis seems to make some good points of the observations.

**Weaknesses:**

- The proposed approach has only been tested on some early architectures. It is not clear how this can be generalized to other models and datasets?
- It is also not clear if the proposed approach is sensitive to different activation functions.
- The comparison between baselines seems to be quite limited to only a few approaches.

**Questions:**

* Can this approach work on other types of architectures like ViT?
* How it may perform with different activation functions?

---

> ### Author Response · Authors · 2024-11-22
> **Response**
>
> ## Responses to Reviewer Comments
>
> ### Reviewer Comment 1:
> “It is also not clear if the proposed approach is sensitive to different activation functions.”
>
> ### Response
> This can easily be dealt with (which we will make clearer in an updated version of the manuscript) by applying the activation functions to extracted activations before applying MIPP.
>
>
> ### Reviewer Comment 2:
> “The comparison between baselines seems to be quite limited to only a few approaches.”
>
> ### Response
> We thank the reviewer for this comment and ask them to suggest further baselines that would complement the current experiments. Our selection criterion was to choose comparators that are representative of a class of systems, which have been thoroughly evaluated and cross-tested by the research community and in industry.
>
> We would like to point out that the key papers in the areas (like those we cite in our paper have essentially the same number of reviews as ours, or less).
>
>
>
> ### Reviewer Comment 3:
> “Can this approach work on other types of architectures like ViT?”
>
> ### Response
> The method is definitely applicable to ViT. In fact, there are no specific 'building blocks' to which our work cannot be applied.

---

### Official Review · Reviewer_3cVS · 2024-11-05

**Soundness:** 3
**Presentation:** 2
**Contribution:** 2
**Rating:** 5
**Confidence:** 4

**Summary:**

The authors propose MIPP to enable real-time pruning, whole-layer pruning and global re-training guarantees for improving the performance of network pruning. Through comprehensive experimental evaluation, they demonstrate that MIPP can effectively prune networks.

**Strengths:**

The authors provide a detailed analysis on the motivation and how the method works, especially on how the mutual information is preserved. And they conduct a lot of experiments to demonstrate the effectiveness of their methods.

**Weaknesses:**

1) Experimental settings and result presentation is not clear.
 - What is untrained network, trained network, pretrained network meaning in Figure 2 and Figure 4?
 - What is LC in Figure 3?
 - In Figure 2, for column 3 and 4, seems the proposed method does not perform well significantly than the baselines.
 - Besides, the latest baseline is year 2022, is there any recent works in 2023 or 2024 to compare?

2) in Figure 4, to my best knowledge, state-of-the-art ResNet50 for ImageNet achieves about 76% accuracy however the proposed approaches can achieve nearly 88% accuracy. Can you explain the settings in detail and what is the percentage of parameters reduced and what is the MACs reduced?

In summary, it is quite unclear how is the comparisons between SOTA and proposed approaches. Besides, some common metrics in comparisons are missing, e.g., FLOPS, #params, MACs. Besides, the baselines seems out-dated.  If the authors could address my concern, I can improve my rating.

**Questions:**

As mentioned in Weakness, I have posed concrete questions for authors. Thanks.

---

> ### Author Response · Authors · 2024-11-22
> **Response**
>
> ## Responses to Reviewer Comments
> ### Reviewer Comment 1:
> “What is untrained network, trained network, pretrained network meaning in Figure 2 and Figure 4?”
>
> ### Response
> In Figure 4, the graphs titled “pre-trained” show the results when applying each method to a trained vision system. On the other hand, “un-trained” implies the application of MIPP and the baselines to a vision system yet to be trained.
>
>
> ### Reviewer Comment 2:
> “What is LC in Figure 3?”
>
> ### Response
> LC stands for “layer collapse”; we will make this clearer in an updated version of the manuscript.
>
>
> ### Reviewer Comment 3:
> “In Figure 2, for column 3 and 4, seems the proposed method does not perform well significantly than the baselines?”
>
> ### Response
> Our method is often outperformed by a re-initialization baseline at high sparsity levels (as are all current pruning methods), and occasionally by GraSP. However, we still demonstrate consistently effective pruning, despite MIPP not being designed for application to an untrained model.
>
>
> ### Reviewer Comment 4:
> “Besides, the latest baseline is year 2022, is there any recent works in 2023 or 2024 to compare?”
>
> ### Response
> We thank the reviewer for their comment. In our opinion, there have not been significant works in this space in 2023/2024. More generally, our selection criterion was to choose comparators that are representative of a class of systems, which have been thoroughly evaluated and cross-tested by the research community, without considering solutions based on their variations.
>
> We would appreciate it if the reviewer could suggest baselines published in 2023 and 2024 they believe would best complement the existing performance evaluation.
>
>
> ### Reviewer Comment 5:
> “In Figure 4, to my best knowledge, state-of-the-art ResNet50 for ImageNet achieves about 76% accuracy however the proposed approaches can achieve nearly 88% accuracy. Can you explain the settings in detail and what is the percentage of parameters reduced and what is the MACs reduced?
>
> ### Response
> We believe there may have been a misunderstanding regarding the graph. Our training accuracies reached nearly 88%, while our test accuracies were below 70%. The pruning ratios are detailed in the figure caption.

---

> > ### Comment · Reviewer_3cVS · 2024-12-02
> > **extra concerns**
> >
> > As this work is mainly to prune the network, some common metrics in comparisons are missing, e.g., FLOPS, #params, MACs. Can you provide more data what are the results of these metrics? It would be nice if a method maintains accuracy while prunes a lot of neurons or saving a lot of computation costs.
> > Say, if your method's accuracy is higher but didn't prune a lot may not indicate your method is good.

---

### Meta-Review · Area_Chair_Fv9c · 2024-12-23

**Metareview:**

The submission proposes Mutual Information Preserving Pruning (MIPP), a method to prune filters/nodes in neural networks that aims to preserve the mutual information between adjacent layers in a neural network.
After the inital round of reviews, this submission received scores of 5, 5, 3, 3. The issues raised by the reviewers are summarized in the section below, and were found to be valid by the ACs.
After the rebuttal and discussion, the reviewers remained unconvinced.
The ACs did not find sufficient reason to overturn the negative consensus.

**Additional Comments On Reviewer Discussion:**

Key among the weaknesses highlighted by the reviewers include:
- Lack of results on the current SoTA architectures, including Vision Transformers and large datasets such as ImageNet.
- Lack of comparison of FLOPs and timing metrics after pruning.

During the rebuttal, the authors did not provide the requested metrics.
The authors stated that they could only compare one baseline on ImageNet due to the lack of computational resources. This is unfortunate since a lot of the numbers reported in the submission are on smaller and simpler datasets such as MNIST, and CIFAR-10/100. While comparisons on these datasets made sense in the past, they are not very representative of real workloads anymore.

---

### Decision · Program_Chairs · 2025-01-22

Reject